# Using Polymer–Surfactant Charge Ratio to Control Synergistic Flocculation of Anionic Particulate Dispersions

**DOI:** 10.3390/polym14173504

**Published:** 2022-08-26

**Authors:** Christopher Hill, Wasiu Abdullahi, Martin Crossman, Peter Charles Griffiths

**Affiliations:** 1School of Science, Faculty of Engineering and Science, University of Greenwich, Chatham Maritime, Kent ME4 4TB, UK; 2Unilever Research, Port Sunlight, Quarry Road East, Bebington, Wirral CH63 3JW, UK

**Keywords:** solvent relaxation NMR, polymer–surfactant interactions, phase separation, adsorption

## Abstract

This study investigates the flocculation induced destabilization of particulate dispersions by oppositely charged polymer–surfactant complexes, with a particular focus on controlling interactions by modulating the charge ratio *Z*, (where *Z* = [+_polymer_]/[−_surfactant_]) via [−_surfactant_] at fixed *C*_polymer_. Cationic hydroxyethyl cellulose (cat-HEC) polymer-sodium dodecylsulfate (SDS) complexes were prepared with either excess polymer (*Z* > 1) or surfactant (*Z* < 1) charges. Anionic particulate dispersions (Ludox and polystyrene-butadiene Latex) were then exposed to the complexes, and solvent relaxation NMR was used to characterize the particle surfaces before and after exposure. In both particulate dispersions, flocculation induced destabilization was enhanced after exposure to cat-HEC-SDS complexes with *Z* > 1, leaving any excess particle surfaces uncoated after gentle centrifugation. However, complexes with *Z* < 1 showed no adsorption and destabilization in the Ludox dispersions and only slight destabilization in the Latex dispersions due to possible hydrophobic interactions. Substituting SDS for non-ionic surfactant (C_12_E_6_) showed no additional destabilization of the dispersions, but post-centrifugation relaxation rates indicated preferential adsorption of C_12_E_6_ onto the particle surfaces. Since the dominant forces are electrostatic, this study highlights the possibility of controlling the interactions between oppositely charged polymer–surfactant complexes and particle surfaces by modulating *Z* through [−_surfactant_].

## 1. Introduction 

Mixtures of polymers and surfactants are ubiquitous in commercially formulated products with applications in detergency, personal care, and drug delivery [1,2,3,4,5,6,7]. Oppositely charged polymer–surfactant mixtures are well known to demonstrate strong synergistic interactions, leading to a rich, tunable, and well-understood phase behaviour. Control over phase behaviour and resulting physicochemical properties can be achieved through adjustments to specific parameters such as but not limited to charge ratio *Z* (where *Z* = [+_polymer_]/[−_surfactant_]), total bulk concentration, polymer molecular weight *M*_w_ (and distribution) and charge density along the polymer backbone (e.g., controlled through adjustments to pH) [2,8,9,10]. Despite the wealth of knowledge around the bulk behaviour of oppositely charged polymer–surfactant complexes, there remains a lacuna in the scientific literature regarding how these complexes interact and adsorb onto different surfaces. In applications such as detergency and conditioning where adsorption is required for performance [11,12,13], this area of research is of significant technical interest, and new knowledge is necessary to make these systems effective and efficient. 

The adsorption of neutral homopolymers and copolymers onto surfaces is reasonably well understood. Many techniques such as solvent relaxation NMR [14], ellipsometry [15] and small-angle neutron scattering [16] have shown that the amount of adsorbed polymer is driven by a complex and subtle interplay between molecular weight and concentration-dependent enthalpic and entropic factors, which in turn drives the polymer conformation at the particle surface [17,18]. Adsorption of poly(ethylene oxide) (PEO) or poly(vinylpyrrolidone) (PVP) onto silica surfaces has become a well explored model system. The consensus mechanism for both neutral polymers involves the formation of hydrogen bonds between surface silanol groups and oxygen atoms on the polymer, thus leading to direct adsorption sites upon the silica within appropriate pH ranges [14]. With the addition of surfactant molecules into such systems, there is the potential for competitive interactions with polymers or particles. Many studies have relied on solvent relaxation NMR to characterize such systems. Mears et al. investigated complex mixtures containing PEO, anionic surfactant (SDS), and negatively charged silica particles [19]. Pre-adsorbed polymer layers were shown to desorb from the silica surface upon addition of between 0–1 wt% SDS, as evidenced by a reduction in the specific relaxation of the solvent within the system to values commensurate with uncoated silica particles and backed up by an observed reduction in the size of the bound polymer layer. Reports on cationic particles such as TiO_2_, have shown that neutral PVP adsorbs relatively weakly to the particle surfaces, but SDS addition below the critical micelle contractions enhances PVP adsorption onto the particle [20]. 

When considering oppositely charged polymer–surfactant mixtures and their interactions with particles, electrostatic interactions must be considered. Terada et al. studied the adsorption of a range of cationic hydroxyethyl celluloses (cat-HEC) with similar charge densities but different molecular weights, and a hydrophobically modified cationic cellulose with a relatively low charge density onto silica surfaces using null ellipsometry [21]. It was shown that molecular weight did not influence the adsorption of this polymer series, however, reductions in charge density were significant, indicating that electrostatics are the driving force for adsorption within these systems. With addition of SDS, associative binding of the surfactant to the oppositely charged polymers defined the interfacial behaviour. Maximum adsorption of the complexes was observed in the low-surfactant concentration (*Z* > 1) one-phase region, whilst higher SDS concentrations led to desorption of the complex from the surface.

More recently, Abdullahi et al. used solvent relaxation NMR to study the interaction between a range of cat-HEC polymers with differing degrees of charge modification with anionic particles in the presence of SDS and mixtures with non-ionic surfactant hexaethylene glycol monododecyl ether (C_12_E_6_) [22]. A universal relationship was shown between initial polymer concentration, degree of charge modification and the mass (surface area) of anionic colloidal material (silica or Latex) removed from the solution. In all systems, cationic-polymer-induced aggregation of the anionic particulate dispersions lead to complete charge neutralization, leaving any remaining particle surfaces uncoated, i.e., a coexisting dense phase of aggregated material and an equilibrium phase of bare particles. The same universality existed in systems containing anionic surfactants, with the appropriate modulation of the phase behaviour being understandable in terms of the stoichiometric binding of surfactant to polymer, emphasizing the domination of electrostatic interactions within oppositely charged particulate dispersions. 

Herein we build upon previous results [22,23] to investigate competitive interactions between cat-HEC-SDS complexes with different particle surfaces, these being silica (hydrophilic) and Latex (hydrophobic). Interactions are explored through the charge ratio (*Z*) phase space, which will be altered through appropriate adjustments to [SDS] at fixed C_polymer_. By taking advantage of the unique ability of solvent relaxation NMR to discriminate between highly mobile (liquid like) and immobile (solid like) protons based on magnetic relaxation rates, we aim to explore the possibility of controlling adsorption and macroscopic phase separation of oppositely charged polymer–surfactant complexes within real-world particulate dispersions. 

## 2. Materials and Methods

### 2.1. Materials

Quaternised hydroxyethyl cellulose polymers have a cellulose backbone and may be regarded as polymeric, quaternary ammonium salts of hydroxyethyl cellulose that have been reacted with trimethylammonium-substituted epoxide (structure shown in Figure 1). The degree of cationic substitution refers to the amount of trimethylammonium substitution along the polymer backbone and, in this study, is expressed in terms of the percentage nitrogen content (*n*). All work here focuses on the cat-HEC polymer with *n* = 0.95% ± 0.15, which Dow Chemical Company kindly supplied. The estimated molecular weight of this cat-HEC polymer is 500,000 g mol^−1^, and a recent small-angle neutron scattering (SANS) analysis [24] showed that the polymer exhibits a rodlike solution conformation in solution, with each unit within the rod exhibiting a dimension R = 0.8 nm and L = 1 nm, i.e., volume = 2–3 nm^3^. Self-diffusion behaviour [25] indicates an effective hydrodynamic (spherical) radius of roughly 25 nm.

Sodium dodecylsulfate (SDS) (Sigma Aldrich, Gillingham, UK, >99.0%) and hexethylene glycol monododecyl ether (C_12_E_6_) (Sigma Aldrich, Gillingham, UK, ≥98%) were used as received. Ludox TM-50 (Sigma Aldrich, Gillingham, UK) is supplied as an aqueous dispersion (50 wt%) and has an average particle diameter of 22 (+/–2) nm), ζ-potential of −45 (+/–1.0) mV (Malvern Zetasizer Nano ZS) and BET surface area (Micromeritics Gemini V 2380 surface area analyzer) of 119 m^2^ g^−1^ [22]. The polystyrene-butylacrylate Latex dispersion with a 2 wt% methacrylic acid content was provided by AkzoNobel and had a particle diameter of 139 nm, ζ-potential of −46 (+/–1) mV and surface area 45 m^2^ g^−1^. All samples were prepared in ultra-pure water with a minimum resistivity of 18.2 MΩ/cm (Purite Select Fusion 80 water purification system). 

### 2.2. Methods-Solvent Relaxation NMR

Spin-spin relaxation time measurements were performed on a Xigo Nanotools Acorn Area low field (13 MHz) spectrometer using the Carr-Purcell-Meiboom-Gill (CPMG) pulse sequence [27,28]. Sample volumes of 0.5 mL were used consistently, and temperatures were maintained at 25 ± 0.5 °C using a circulating water bath. Typically, 718 data points (echo cycles) were collected for each scan. The signal was averaged over four scans for each sample with an interpulse spacing of 0.50 ms between the 90° and 180° pulses. A recycle delay of about 5*T*_1_ was allowed between each cycle to allow full recovery of the magnetization between acquisitions. Three concordant *T*_2_ values were measured for each sample, the results, and standard deviations of these were used to calculate the error bars for each measurement. The Acorn AreaQuant software was used to fit all the relaxation decay curves measured to a single exponential decay using Equation:(1)$$M_{y}\left(t\right)=M_{y}\left(0\right)e^{t/T_{2}}$$
where *M_y_*(0) is the transverse magnetization immediately after the 90° pulse.

When a water molecule is constrained at a surface, its dynamics become anisotropic or restricted, such that the efficiency of dipolar relaxation is increased, and the (spin-spin) relaxation time *T*_2_ is shortened [14]. The overall relaxation rate in the fast exchange limit between ‘free’ water molecules with a relaxation time *T*_2f_ and ‘bound’ molecules with a relaxation time *T*_2b_ is then given by:(2)$$\frac{1}{T_{2}}=\frac{\left(1-p_{b}\right)}{T_{2f}}+\frac{p_{b}}{T_{2b}}$$
where *p_b_* is the time-averaged probability of finding a randomly chosen solvent molecule at the particle surface. The relaxation time is converted into a rate $\frac{1}{T_{2}}=R_{2}$ and normalized by the relaxation rate of the water employed to prepare the dispersion, $R_{2}^{0}$, to yield the specific relaxation rate *R*_2sp_:(3)$$R_{2\mathrm{sp}}=\frac{R_{2}}{R_{2}^{0}}-1$$

In the absence of polymer or surfactant, the *R*_2sp_ for a solvent in a particulate dispersion will scale linearly with the available particle surface area (*σ*) due to the increase in *p_b_*, and tends to zero in the absence of (any) surface. Solvent molecules associated with free polymers or within polymer regions away from the near-surface region, i.e., loops and tails, show no significant change in relaxation rate due to high mobility. However, when adsorbed as a train layer at the particle surface, an enhancement in the overall relaxation rate is observed due to an increased proportion and/or residence time of water molecules in the near-surface regions [14]. 

## 3. Results and Discussion

### 3.1. Particle Only Systems

In deionized water, the unconstrained mobility of the molecules leads to a relatively long spin-spin relaxation time (*T*_2_) of ~2400 ms at 25 °C and 13 MHz. As mentioned above, increasing the concentration (surface area) of particles leads to a reduction in *T*_2_ and a universal linear relationship between relaxation rate (1/*T*_2_) and particle concentration is invariably observed [14]. The specific relaxation rate vs. particle concentration (surface area) for the aqueous particulate dispersions used in this work are presented below in Figure 2. 

The data in Figure 2 display a linear dependence of relaxation rate (*R*_2sp_) with increasing particle concentration (surface area), indicating an absence of any form of aggregation, adsorption or flocculation within the studied concentration range. This linear dependence can be predicted by Equation (2) for systems with a fast exchange between water molecules associated with the surface and those in the bulk solution [29,30]. 

The data indicate that for a given particle concentration, Latex particles have an increased relaxation rate enhancement when compared to Ludox, $R_{2\mathrm{sp}}^{\mathrm{enh}\;\left(\mathrm{Latex}\right)}=0.30\;\mathrm{wt}{\%}^{-1}$ and $R_{2\mathrm{sp}}^{\mathrm{enh}\;\left(\mathrm{Ludox}\right)}=0.25\;\mathrm{wt}{\%}^{-1}$, respectively. Relaxation rate enhancement differences between particle types result from the size and specific surface area of the particles and the nature and chemical characteristics of the surface [31]. For example, in aqueous-based particulate dispersions, surfaces with hydrophilic and wetting characteristics can produce higher relaxation rates than hydrophobic particles. Here, the polystyrene-butylacrylate Latex particles are hydrophobic in nature but are stabilized by methacrylic acid, effectively producing sites for favourable interactions with water molecules. The possibility of controlling the hydrophobicity of Latex particles through adjustments to the amount of acid stabilizer used in the synthetic procedure has been successfully demonstrated [32,33]. Beshah et al. showed that the presence of acid groups on Latex particles leads to a significant reduction in the adsorption of hydrophobic end groups of modified ethylene oxide urethane polymers compared to more hydrophobic Latex particles with fewer acid groups.

Previous relaxation NMR studies on Latex and Ludox particles have reported relaxation rate enhancement values of 0.24 wt%^−1^ and 0.50 wt%^−1^ for Ludox and Latex (4 wt% methacrylic acid stabilization), respectively [22,23]. Therefore, results reported here are in keeping with previous literature and highlight how the presence of acid groups on hydrophobic Latex particles can present relaxation rate enhancements greater than hydrophilic Ludox particles.

### 3.2. Competitive Interactions within Particulate Based Dispersions–Oppositely Charged Polymer–Surfactant Complexes and Silica

Order of addition is known to play a key role in the adsorption behaviour of polymers and polymer–surfactant complexes onto particle surfaces [21,22,34,35]. This study aims to investigate the effect that charge ratio (*Z*) (where *Z* = [+_polymer_]/[−_surfactant_]) has on the adsorption of oppositely charged polymer–surfactant complexes onto particles. Adjustments to *Z* were made through changes to [SDS], and the cat-HEC polymer concentration remained constant at 0.1 wt%. All samples were prepared by first mixing the cat-HEC polymer and surfactant before adding particles. Interactions between cat-HEC polymers and SDS are strong and synergistic [2,24,25], and previous reports have highlighted the preference for cat-HEC-SDS interactions over cat-HEC-Ludox [21,22]. Before relaxation NMR and dry weights characterization, samples were gently centrifuged (5000 RPM for 10 min, 130 N) following previously reported experimental protocols that yielded a dense phase containing the flocculated species and a coexisting stable dispersion [22]. All characterization was carried out on the stable dispersion, assuming that the bulk of the weight of the dried sample will arise from the particles (we term this the ”equilibrium particle concentration”). 

Data are presented in terms of relaxation rate enhancements (*R*_2sp_) as a function of the equilibrium particle concentration, determined through dry weight analyses. Another simple presentation of the data is to compare the initial and equilibrium mass fractions post exposure to the oppositely charged polymer–surfactant complex and centrifugation protocol to quantify the amount of solid material that is rendered unstable [23]. Figure 3 shows these data for the mixtures containing cat-HEC polymer (*n* = 0.95%), SDS and Ludox. 

Making the reasonable assumption that any cat-HEC polymer, SDS or polymer–surfactant complexes that adsorbs to the surface would perturb *R*_2sp_ [14,22,23,36], the alignment of all data sets with the linear regression model of the Ludox-only data in Figure 3a strongly suggests that the particles detected in the solvent relaxation experiment are uncoated. These results align with previous reports that have demonstrated the domination of cat-HEC polymer-SDS interactions over polymer-particle interactions [21,22]. 

There are two scenarios in which particles may remain uncoated after exposure to the cat-HEC polymer-SDS complexes. The first is that interactions between the complex and particles lead to flocculation; the flocculated particles are then removed through the centrifugation process, thus leaving any stable particles uncoated. Abdullahi et al. demonstrated this with a range of cat-HEC polymers with minor perturbations in the charge density, with results highlighting the increased capacity of the charged polymers to promote polymer bridging flocculation of the silica when compared to uncharged analogues as a result of the electrostatic interaction [22]. The second scenario in which the particles could remain uncoated is that there is no interaction between the particles and the cat-HEC polymer-SDS complexes, hence destabilization through the flocculation mechanism will not occur. This second scenario will be significantly influenced by Z, which can be explored simply through adjustments to [SDS] to render the complexes with an effective positive or negative charge. Polymer–surfactant complexes with a charge ratio in favour of the positively charged polymer (i.e., *Z* > 1) will invariably promote destabilization and phase separation would be expected. On the other hand, when *Z* < 1 (excess of surfactant charges), the additional surfactant can dissolve the precipitate formed at charge neutrality (*Z* = 1) and stabilize it colloidally. In this situation, one has a micellar solution in which the polymer is fully complexed and stabilized by the surfactants, meaning that the effective charge of the polymer–surfactant complex will be negative and the capacity for flocculation induced destabilization is likely to be significantly reduced or even fully ‘turned off’ (see Graphical Abstract).

The data in Figure 3b allows for a comparison between the equilibrium and initial particle concentrations to provide insight into the mass percentage of the particles rendered unstable after exposure to the polymer–surfactant complexes with different *Z* values. It should be emphasized that there is no loss of particles during the centrifugation protocol in the particle-only data, hence any observed deviations from the X = Y guide to the eye in Figure 3b data sets will indicate loss of particles as a result of cat-HEC polymer-SDS complex destabilization. 

For cat-HEC polymer-SDS complexes with *Z* < 1 (excess surfactant charges), all data follow the X = Y linear relationship and there is no observed particle loss due to destabilization. Since the interactions within these systems tend to be dominated by electrostatics [21,22], it is reasonable to assert that the effective negative charge on the cat-HEC polymer- surfactant complexes at *Z* < 1 provides sufficient repulsive interactions with the anionic silica particles and the capacity for flocculation induced destabilization is removed. Similar behaviour has been demonstrated in mixtures containing anionic SDS micelles and anionic silica nanoparticles [37] and mixtures containing oppositely charged polymer–surfactant complexes and anionic silica [10,38]. Contrast variation small-angle neutron scattering was used to systematically match out the scattering contribution from SDS micelles and silica nanoparticles, respectively, and results highlighted that the like charges on the surfactants and particles led to the presence of coexisting micelles and nanoparticles and absence of any physical interaction between the two components [37]. Mohr et al. used ellipsometry to study cationic copolymers, vinylpyrrolidone and quaternized vinylimidazol, with SDS at the silica-aqueous interface [38]. Results showed that adsorption of the complexes onto the silica peaked at SDS concentrations that corresponded closely to the concentration of cationic charges on the copolymer, and a significant decrease in adsorption was observed at high concentrations of SDS as a result of excess SDS binding to the complexes. 

Considering the data for the cat-HEC polymer–surfactant complex with *Z* > 1 (excess cationic polymer charges) and the cat-HEC/silica mixture in Figure 3b. Both data sets show a clear deviation from the X = Y line, indicating a loss of particles due to interactions with the cationically charged species. The cat-HEC polymer alone (down triangles) shows a greater removal of silica from the dispersion than the cat-HEC polymer-SDS complex (*Z* = 1.2). As per the hypothesis of this work, this is a result of the preferential binding of the SDS to the cat-HEC polymer, which reduces the cationic charge density on the polymer thereby reducing the concentration of silica that can be removed from the dispersion compared to the polymer alone. If one considers the data that display significant reductions in the equilibrium particle concentrations, the concentration of particles removed from the dispersions are 1.2 ± 0.4 wt% and 2.1 ± 0.8 wt% for the cat-HEC polymer–surfactant complex and cat-HEC polymer alone, respectively. The recent investigation by Abdullahi et al. reported a consistent removal of 3 wt% in mixtures containing cat-HEC polymer LR (*n* = ~0.9%) and silica, this being comparable (within experimental error) with the 2.1 ± 0.8 wt% value reported in this work [23]. 

### 3.3. Competitive Interactions within Particulate Based Dispersions–Oppositely Charged Polymer–Surfactant Complexes and Latex

To explore how the possibility of hydrophobic interactions affects the adsorption and subsequent destabilization of a particulate dispersion, experiments were carried out on Latex particles. The same experimental procedure was followed regarding the cat-HEC polymer-SDS complexes investigated and the order of addition, however the parameters used during the centrifugation step were adjusted as per previous investigations to ensure no particle loss (2000 RPM for 5 min, 21 N) [22,23]. The data for the mixtures containing cat -HEC polymer (*n* = 0.95%), SDS and Latex are presented in Figure 4. 

Figure 4a display the relaxation rate enhancements as a function of equilibrium particle concentration after exposure to the different cat-HEC polymer-SDS complexes. As with the Ludox data presented in Figure 3a, the observed linearity in all Latex datasets shows that the remaining particles within the stable dispersion are uncoated. Any particles that interact with the cat-HEC polymer-SDS complexes are therefore rendered unstable and are subsequently removed from the solution during centrifugation.

Latex particles have the capability of interacting via both hydrophobic [32,33,39] and electrostatic interactions [22,23,39,40], hence flocculation induced destabilization may be enhanced in systems that display both of these capabilities. Figure 4b presents the data for equilibrium particle concentration as a function of initial particle concentration, showing the mass percentage of Latex particles rendered unstable after exposure to the cat-HEC polymer-SDS complexes with different *Z* values. All data in Figure 4b show a clear deviation from the X = Y guide to the eye, indicating a loss of particles after exposure to the different cat-HEC polymer-SDS complexes. A clear linear offset from the X = Y guide is shown in these data, demonstrating that at each studied *Z* value, the rate of particle loss in these systems is comparable. It is important to note the relationship between the mass of particles removed from the dispersion and the *Z* value of the polymer–surfactant complex. The complex with the lowest *Z* value (excess surfactant charges) removes 0.7 ± 0.2 wt% of particles compared to 3.0 ± 1.0 wt% removed by the highest *Z* (excess cationic polymer charges). This relationship emphasizes that the dominant force destabilizing these systems is an electrostatic one, such that complexes with a higher degree of cationic character interact with, and destabilize more, Latex particles. 

As previously discussed, the hydrophobicity of Latex particles can be controlled through adjustments to the amount of (methacrylic acid) stabilizer used in the synthetic procedure [32,33]. The modest 2 wt% methacrylic acid stabilizer present in the Latex particles here has been shown to be low enough to allow for hydrophobic interactions with polymers [33]. For the cat-HEC polymer used in this study, hydrophobic interactions are possible via the side chain modification but not the polymer backbone, as shown through small-angle neutron scattering studies [24]. Of course, SDS surfactant “tails” may also contribute to hydrophobic interactions and reports have highlighted this with Latex particles [39]. 

Ludox containing systems showed no particle loss after exposure to the complexes that carry an effective negative charge (*Z <* 1) because of sufficient repulsive interactions (see Figure 3b). Since the Latex particles also carry a negative charge (ζ-potenatial = −46 ± 1 mv), the observed particle destabilization after exposure to *Z* < 1 complexes must occur via hydrophobic interactions. The larger particle diameter of Latex compared to Ludox 139 nm and 22 nm, respectively, may also increase any flocculation induced destabilization which would be expedited by the presence of adsorbed charged species. 

Overall, it has been shown that electrostatic interactions strongly dominate the flocculation induced destabilization in anionic Ludox and methacrylic acid stabilized Latex particulate dispersions. By making simple adjustments to the charge ratio of the pre-formed cat-HEC polymer-SDS complexes through adjustments to the [SDS], the amount of particulate material removed from solution (adsorbed) can be controlled. When considering these results in a real-world context whereby Ludox and Latex particles are proxies for hair/fibre and dirt surfaces, respectively, the *Z* value of the polymer–surfactant complex must be carefully considered. For a sufficient conditioning process to occur, the complex must remain adsorbed onto the hair/fiber surfaces whilst removing any dirt particles. Results suggest that complexes with *Z* > 1 demonstrate strong interactions with both particle types and that additional hydrophobic interactions allow for enhanced interactions, indicating that simultaneous conditioning and cleaning should occur with these complexes. However, to understand these interactions in a real-world context, any preferential interactions between the particle surfaces and polymer–surfactant complexes must be probed in binary particulate dispersions. Solvent relaxation NMR has been shown to be a viable method for characterising preferential adsorption of cationic polymers within binary particulate dispersions [23], hence studying interactions between particle surfaces and polymer–surfactant complexes in binary particulate dispersions is the next logical step. 

### 3.4. Competitive Interactions–Non-Ionic Surfactant Effects

Next, mixtures containing non-ionic surfactant (C_12_E_6_), cat-HEC polymer and particles are investigated. In this case, there will be no preferential electrostatic interactions between the cat-HEC polymer and surfactant within these mixtures, instead non-ionic surfactant adsorption to the particle surfaces may be promoted. As with the previous experiments, the cat-HEC polymers (0.1 wt%) were mixed with 0.5 × critical micelle concentration (CMC) C_12_E_6_ prior to exposure to the Ludox and Latex particles. The same centrifugation process was carried out. Data for both Ludox and Latex are presented below in Figure 5.

Considering the relaxation rate enhancement data for both particle systems in Figure 5a,c. As previously shown for the cat-HEC polymer/particle data, the relaxation rate enhancement is in line with the particle only data, indicating that the particles remaining after the centrifugation process are uncoated. However, a slight increase in *R*_2sp_ and corresponding gradient change is observed for both particles after exposure to cat-HEC-C_12_E_6_ mixtures. This increase in *R*_2sp_ would indicate an increase in *p_b_* as per Equation (2), i.e., more bound water molecules are at the particle surfaces. Since these systems destabilize through electrostatic induced flocculation [21,22,23], the observed *R*_2sp_ enhancements after exposure to the cat-HEC-C_12_E_6_ mixtures must be a result of adsorption of the non-ionic surfactant molecules onto the particle surfaces. This behaviour is somewhat expected since non-ionic surfactants are known to adsorb onto Ludox [37,41] and Latex [42,43] surfaces. Furthermore, relaxation rate enhancements after exposure of particle surfaces to non-ionic surfactants have been previously reported, resulting from increased water association with surfactant headgroups on the particle surfaces [44].

Interestingly, the data do not indicate any significant increase in the destabilization of the particulate dispersions after exposure to the cat-HEC-C_12_E_6_ mixtures compared to the cat-HEC polymer alone. The data in Figure 5b,d show the equilibrium vs. initial particle concentrations after exposure of the particles to cat-HEC only and cat-HEC-C_12_E_6_ mixtures. For each particle system, there is good overlap between the two different data sets within experimental error. Ludox systems show a 1.6 ± 1 wt% and 1.4 ± 1 wt% particle loss after exposure to the cat-HEC only and cat-HEC-C_12_E_6_ mixture, respectively. The Latex systems show a 3.3 ± 1 wt% and 3.1 ± 1 wt% particle loss after exposure to the cat-HEC only and cat-HEC-C_12_E_6_ mixture, respectively. These comparable data within the particle systems emphasize that the destabilization process is fundamentally controlled through electrostatics and that the non-ionic surfactants do not facilitate any flocculation induced destabilization. Instead, the data indicate that the non-ionic surfactants show an increased tendency to adsorb onto the particle surfaces as opposed to the cat-HEC polymers, remaining adsorbed even after the centrifugation procedure (see Graphical Abstract).

## 4. Conclusions

Solvent relaxation NMR has been used to investigate competitive interactions between oppositely charged cationic hydroxyethyl cellulose polymers (*n* = 0.95%)-sodium dodecylsulfate (SDS) complexes and particulate dispersions (Ludox and Latex). This study aimed to determine whether it is possible to control the adsorption and associated flocculation induced destabilization by modulating the charge ratio *Z* (where *Z* = [+_polymer_]/[−_surfactant_]) of oppositely charged complexes via [SDS].

Ludox (ζ-potential = −45 mV) and Latex (ζ-potential = −46 mV) particle dispersions demonstrated expected relaxation rate behaviour, with no indications of any form of aggregation, adsorption, or flocculation within the studied concentration range. The polystyrene-butylacrylate Latex showed larger relaxation rate enhancements when compared to Ludox, this being a result of the 2 wt% methacrylic acid used to stabilize the Latex dispersions which provides sites for favourable interactions with water molecules. Results showed that for a given particle concentration, the relaxation rate enhancements are $R_{2\mathrm{sp}}^{\mathrm{enh}\;\left(\mathrm{Latex}\right)}=0.30\;\mathrm{wt}{\%}^{-1}\;$ and $R_{2\mathrm{sp}}^{\mathrm{enh}\;\left(\mathrm{Ludox}\right)}=0.25\;\mathrm{wt}{\%}^{-1}$, respectively.

In both particulate dispersions, flocculation induced destabilization was promoted after exposure to cat-HEC-SDS complexes with *Z* > 1 (excess cationic charges), leaving any excess particle surfaces uncoated after exposure centrifugation. However, exposure to complexes with *Z* < 1 (excess surfactant charges) showed no adsorption and destabilization in the Ludox dispersions and only slight destabilization in the Latex dispersions due to possible hydrophobic interactions. Interestingly, cat-HEC-non-ionic surfactant mixtures showed no additional destabilization of the dispersions, but post-centrifugation relaxation rates were enhanced due to preferential adsorption of the non-ionic surfactant (C_12_E_6_) onto the particle surfaces. 

The reported relationship between the concentration of particles rendered unstable and the cat-HEC-SDS complexes *Z* values emphasizes that electrostatic interactions strongly dominate the flocculation induced destabilization in anionic Ludox and methacrylic acid stabilized Latex particulate dispersions. Overall, this work has demonstrated that by modulating the charge ratio of the pre-formed cat-HEC polymer-SDS complexes through simple adjustments to the [SDS], the amount of particulate material removed from a solution via adsorption is tunable and easily controlled. 

## Figures and Tables

**Figure 1 polymers-14-03504-f001:**
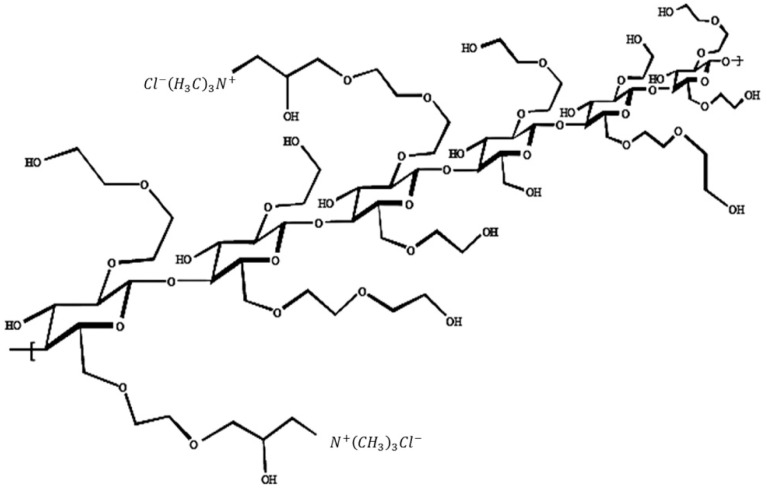
Generic structure of cationic hydroxyethyl cellulose (cat-HEC) polymers. Adapted with permission from Ref. [26]. 2004 Oxford University Press.

**Figure 2 polymers-14-03504-f002:**
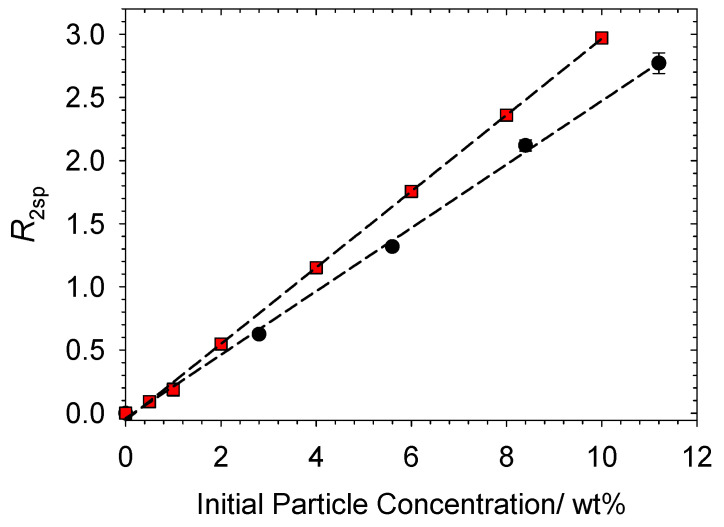
Enhancement in *R*_2sp_ as a function of particle concentration: Symbols: ●, Ludox TM-50; ■, Latex.

**Figure 3 polymers-14-03504-f003:**
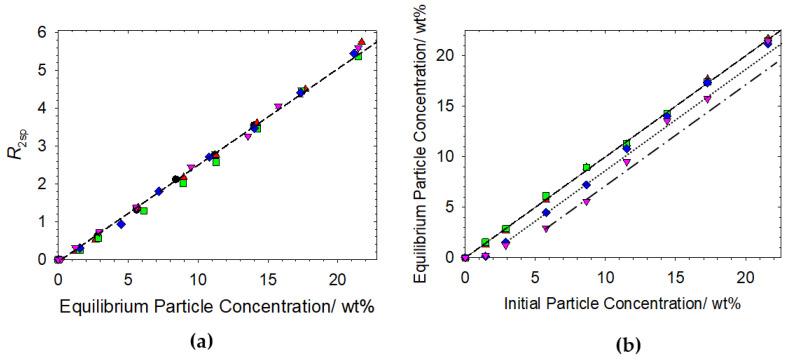
(**a**) Specific relaxation rates as a function of the equilibrium particle concentration for mixtures containing cat-HEC (*n* = 0.95%), SDS and Ludox. Regression plot is through Ludox-only data. (**b**) Equilibrium particle concentration as a function of initial particle concentration for the mixtures. X = Y plot is included as a guide for the eye; dotted and dot dash lines are linear offsets from the X = Y plot. Symbols: ●, Ludox only; ▲, 4 mM SDS + 0.1 wt% cat-HEC polymer (*Z* = 0.15); ■, 2 mM SDS + 0.1 wt% cat-HEC polymer (*Z* = 0.30); **♦**, 0.5 mM SDS + 0.1 wt% cat-HEC polymer (*Z* = 1.20); ▼, 0.1 wt% cat-HEC polymer.

**Figure 4 polymers-14-03504-f004:**
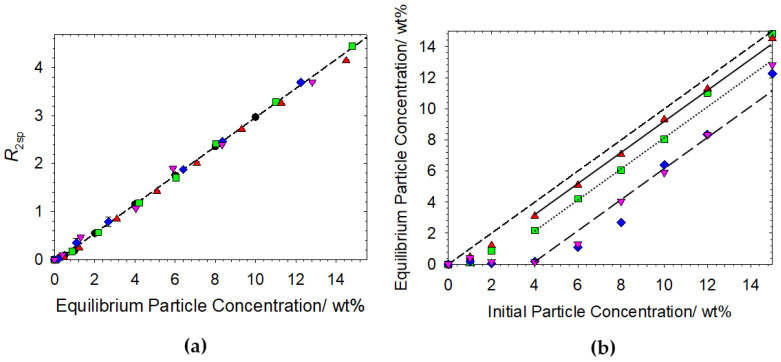
(**a**) Specific relaxation rates as a function of the equilibrium particle concentration for mixtures containing cat-HEC (*n* = 0.95%), SDS and Latex. Regression plot is through Latex-only data. (**b**) Equilibrium particle concentration as a function of initial particle concentration for the mixtures. X = Y plot is included as a guide for the eye; dotted and dot dash lines are linear offsets from the X = Y plot. Symbols: ●, Latex only; ▲, 4 mM SDS + 0.1 wt% cat-HEC polymer (*Z* = 0.15); ■, 2 mM SDS + 0.1 wt% cat-HEC polymer (*Z* = 0.30); ♦, 0.5 mM SDS + 0.1 wt% cat-HEC polymer (*Z* = 1.20); ▼, 0.1 wt% cat-HEC polymer.

**Figure 5 polymers-14-03504-f005:**
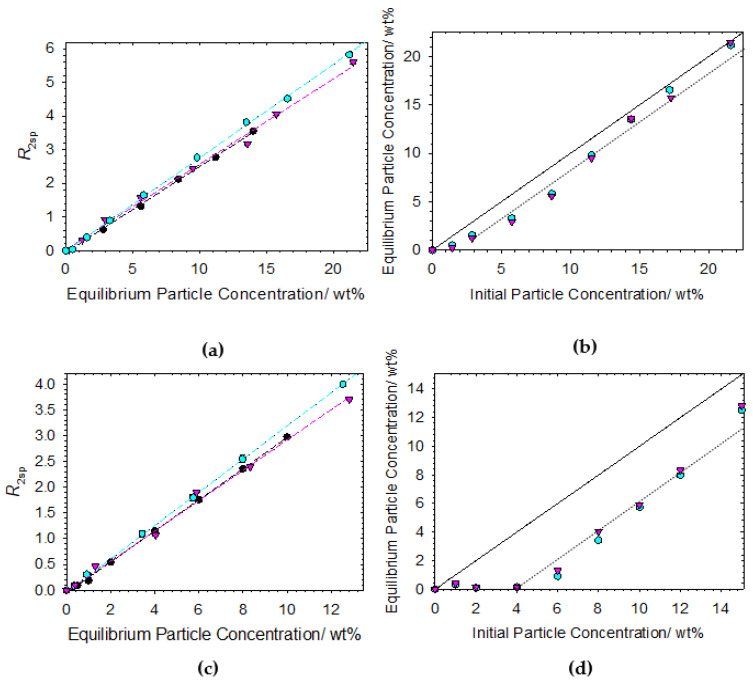
(**a**,**c**) Specific relaxation rates as a function of the equilibrium particle concentration for mixtures containing cat-HEC (*n* = 0.95%), 0.5 × CMC C_12_E_6_ and Ludox (**a**)/Latex (**c**). Regression plot is through each data set. (**b**,**d**) Equilibrium particle concentration as a function of initial particle concentration for the mixtures containing Ludox (**b**) and Latex (**d**). X = Y plots in (**b**,**d**) are included as guides for the eye; dotted lines are linear offsets from the X = Y plot. Symbols: ●, particle only; ⬢, 0.5 × CMC C_12_E_6_ + 0.1 wt% cat-HEC polymer; ▼, 0.1 wt% cat-HEC polymer.

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
