# Peer review of "Using Polymer–Surfactant Charge Ratio to Control Synergistic Flocculation of Anionic Particulate Dispersions"

_polymers, 2022, doi:10.3390/polym14173504_

Round 1

Reviewer 1 Report

The authors reported on the modulation of charge ratio of oppositely charged complex through [SDS] to control the adsorption and associated flocculation induced destabilization.

The results demonstrated that the solvent relaxation NMR technique is powerful for characterizing the aggregation, adsorption and flocculation process. I recommend the publication of this work on Polymers upon the condition that the comments below are well addressed.

[1] The authors claimed that “the larger particle diameter of Latex compared to Ludox 139 nm and 22 nm, respectively, may also increase any flocculation induced destabilization which would be expedited by the presence of adsorbed charged species. Why don’t perform the experiments with the same size of the Ludox and Latex instead? This experiment is critical.

[2] The authors may conduct the water contact angle experiment to characterize the hydrophilic/hydrophobic properties of the Ludox and Latex.

[3] What are the relative centrifugal force and centrifugation time for Ludox and Latex?

[4] In Figure 4b & 5d, why all the particles removed when the initial particle concentration of Latex below 4 wt% when Z>1?

[5] There are some abbreviations not mentioned before such as SANS and CMC.

[6] Why 0.5 x CMC concentration is chosen for C12E6 to study the non-ionic surfactant adsorption effect?

[7] The section for conclusion part should be “4” instead of “5”.

Reviewer 2 Report

Review of paper ‘Using polymer-surfactant charge ratio to control synergistic flocculation of anionic particulate dispersions’ prepared by Christopher Hill, Wasiu Abdullahi, Martin Crossman, and Peter Charles Griffiths1.

Manuscript polymers-1883821 is focused on the presentation of interaction between polymers and surfactants and their use in flocculation systems. I have some suggestions that authors may consider prior to publication of this work:

1. The authors should make it clear whether the proposed research is only exploratory or can be transferred in the future to the estimation of potential flocculation agents or to the selection of suitable agents for potential contamination.

2. Equation 3 requires clarification. If T2 is the relaxation time, that is, with a unit of a second, then R2 has the unit of s-1. In this case, the units on the right-hand side of the equation do not agree (no unit here). Rate usually has the unit.

4. Stable dispersion is the key element here. Can the authors address the issue of disruption to this stability when, for example, electrolytes are added? This is an important aspect for a potential application.

5. Authors wrote. ‘This indicates that the simultaneous conditioning and cleaning processes should occur with these complexes however to understand these interactions in a real-world context fully, further investigations are required to determine any preferential interactions in mixed particulate dispersions’ (lines 386-389). Please add some more information here.

6. Figure 1 is blurred and needs to be corrected.

7. In the description of the Figures it is better to use symbols rather than a description.

8. I don’t see any graphical abstract (see references to GA in line 273).

Round 2

Reviewer 1 Report

It can be accepted in present from.